# Assessing the Anthelmintic Candidates BLK127 and HBK4 for Their Efficacy on *Haemonchus contortus* Adults and Eggs, and Their Hepatotoxicity and Biotransformation

**DOI:** 10.3390/pharmaceutics14040754

**Published:** 2022-03-30

**Authors:** Markéta Zajíčková, Lukáš Prchal, Ivan Vokřál, Linh Thuy Nguyen, Thomas Kurz, Robin Gasser, Klára Bednářová, Magdalena Mičundová, Beate Lungerich, Oliver Michel, Lenka Skálová

**Affiliations:** 1Department of Biochemical Sciences, Faculty of Pharmacy, Charles University, Heyrovského 1203, 50005 Hradec Králové, Czech Republic; zajickm@faf.cuni.cz (M.Z.); nguyenli@faf.cuni.cz (L.T.N.); bednarokl@faf.cuni.cz (K.B.); micundom@faf.cuni.cz (M.M.); 2University Hospital Hradec Kralove, Biomedical Research Centre, Sokolska 581, 50005 Hradec Kralove, Czech Republic; prchal.l@email.cz; 3Department of Pharmacology and Toxicology, Faculty of Pharmacy, Charles University, 50005 Hradec Králové, Czech Republic; vokral@faf.cuni.cz; 4Institute of Pharmaceutical and Medicinal Chemistry, Heinrich-Heine University, 40225 Düsseldorf, Germany; thomas.kurz@hhu.de (T.K.); beate.lungerich@hhu.de (B.L.); oliver.michel@hhu.de (O.M.); 5Department of Veterinary Biosciences, Melbourne Veterinary School, The University of Melbourne, Parkville, VIC 3010, Australia; robinbg@unimelb.edu.au

**Keywords:** ATP, drug development, drug metabolism, drug resistance, nematodes, new anthelmintics

## Abstract

As a widely distributed parasitic nematode of ruminants, *Haemonchus contortus* has become resistant to most anthelmintic classes, there has been a major demand for new compounds against *H. contortus* and related nematodes. Recent phenotypic screening has revealed two compounds, designated as BLK127 and HBK4, that are active against *H. contortus* larvae. The present study was designed to assess the activity of these compounds against *H. contortus* eggs and adults, hepatotoxicity in rats and sheep, as well as biotransformation in *H.* *contortus* adults and the ovine liver. Both compounds exhibited no inhibitory effect on the hatching of eggs. The benzyloxy amide BLK127 significantly decreased the viability of adults in sensitive and resistant strains of *H. contortus* and showed no hepatotoxic effect, even at the highest concentration tested (100 µM). In contrast, HBK4 had no impact on the viability of *H. contortus* adults and exhibited significant hepatotoxicity. Based on these findings, HBK4 was excluded from further studies, while BLK127 seems to be a potential candidate for a new anthelmintic. Consequently, biotransformation of BLK127 was tested in *H. contortus* adults and the ovine liver. In *H. contortus*, several metabolites formed via hydroxylation, hydrolysis and glycosidation were identified, but the extent of biotransformation was low, and the total quantity of the metabolites formed did not differ significantly between the sensitive and resistant strains. In contrast, ovine liver cells metabolized BLK127 more extensively with a glycine conjugate of 4-(pentyloxy)benzoic acid as the main BLK127 metabolite.

## 1. Introduction

*Haemonchus contortus* is one of the most important trichostrongyloid nematodes infecting ruminants; it has a worldwide distribution and causes significant economic losses to livestock industries [1,2]. Although strategies, including grazing management, nutrition supplementation, selective breeding or vaccination are important in the control of haemonchosis, the use of anthelmintics continues to play an important role in effective control [3]. Nevertheless, this and related parasites have developed genetic resistance to all currently used classes of anthelmintics [4], such that there has been a need to find new chemical entities as anthelmitics with distinct mechanisms or modes of action [5].

Drug development is a long-term and costly process, which includes multiple steps. The process starts with the search and selection of “hit” compounds, usually via high-throughput screening (HTS) of compound libraries [5,6] followed by toxicity screening and defining pharmacokinetic and pharmacodynamic parameters [7,8,9]. A recent screen of 236 chemically diverse compounds from the “Kurz-box” (heterocyclic compounds, hydroxamic acid-based metalloenzyme inhibitors, peptidomimetics and various intermediates formed by T. Kurz and B. Lungerich at Heinrich-Heine-University, Düsseldorf) identified two candidates, designated as BLK127 and HBK4, with activity against exsheathed third- (xL3) and fourth- (L4) larvae of *H. contortu**s*. Both BLK127 and HBK4 induced phenotypic changes and inhibited xL3 and L4 motility at seven days of incubation [10]. Compound BLK127 induced an evisceration phenotype, characterized by a protrusion of internal tissues and organs through the excretory pore in xL3s [11], whereas compound HBK4 induced a “curved” phenotype associated with the presence of large vacuoles. Moreover, compound BLK127 significantly inhibited L4 development after seven days of incubation, while HBK4 did not [10]. As the sensitivities of xL3s and L4s to anthelmintics differ from those of other stages, due to their distinct biochemistry and physiology [12,13], other developmental stages, particularly the adults, should be tested [6].

As the liver is the main organ for drug metabolism, drug-induced liver injury often occurs, such that the assessment of potential hepatotoxicity is essential [14]. Several in vitro models have been developed for hepatotoxicity testing, with each model showing both advantages and disadvantages. One of the most widely used models is hepatocytes [15], being amenable to high-throughput screening [16,17]. However, they do not represent all cells and tissues of the liver. For this reason, precision-cut liver slices (PCLS), representing the intact liver model, are often used [17,18].

The metabolite identification can provide information about sites to be blocked or modified, help to predict in vivo situations, and toxicity alters as well as assists in understanding the mechanism of action of the drug [19]. Moreover, biotransformation often leads to drug deactivation, with an acceleration of this process linked to drug resistance [1,4,20,21,22].

The main objectives of this study were therefore the evaluation of the BLK127 and HBK4 efficacy and biotransformation on adults of *H. contortus* from one sensitive to common anthelmintics (ISE, Inbred-Susceptible-Edinburgh, MHco3) and two resistant strains (IRE, Inbred-Resistant-Edinburgh, MHco5; and WR, White River, MHco4) [21], in addition to their ovicidal effects. Moreover, hepatotoxicity of those compounds in sheep and rat liver models (PCLS and isolated hepatocytes) and biotransformation in sheep liver models were also assessed.

## 2. Materials and Methods

### 2.1. Chemicals and Reagents

*N*-(benzyloxy)-4-(pentyloxy)benzamide (BLK127) and *N*-(4-(5-(phenylsulfonamido)-1*H*-benzo[*d*]imidazol-2-yl)phenyl)benzenesulfonamide (HBK4) (Table A1) were synthesized at the Institute of Pharmaceutical and Medicinal Chemistry, Heinrich-Heine-University, Düsseldorf, Germany [10]. *N*-(benzyloxy)-3-ethoxy-2-naphthamide (internal standard-IS) was obtained from Aldrich Market Select (Milwaukee, WI, USA). Acetonitrile (ACN, LC-MS grade), ethanol, ethyl acetate (HPLC grade) and methanol (HPLC-MS grade) were obtained from VWR International Ltd. (Stříbrná Skalice, Czech Republic). The Pierce™ BCA Protein Assay Kit and Williams’ Medium E-GlutaMAX (32551) was purchased from Thermo Fisher Scientific (Prague, Czech Republic). Collagenases (from Clostridium histolyticum, C5138), Roswell Park Memorial Institute 1640 medium (RPMI-1640, R8758), formic acid (LC-MS LiChropurTM, 97.5–98.5%), and all other chemicals were purchased from Sigma-Aldrich (Prague, Czech Republic). Ultra-pure water of the ASTM I type (resistance 18.2 MΩ.cm at 25 °C) was prepared using the Barnstead Smart2Pure 3 UV/UF system (Thermo Fisher Scientific, Bremen, Germany).

### 2.2. Parasites and Hosts

All experimental procedures were approved by the Ethics Committee of the Ministry of Education, Youth and Sports (Protocol MSMT-25908/2019) and performed in accordance with Czech Act No 246/1992 Coll. on the Protection of Animals against Cruelty, as amended.

Adults and eggs of *H. contortus* were produced using an established approach [23]. Helminth-free lambs (6-months-old) were orally infected with 5000–8000 L3s of *H. contortus* (strain ISE, IRE or WR). The infective dose was dependent on the weight of the individual sheep. The eggs were isolated from the feces using a sucrose flotation procedure [23] four weeks after infection. The *H. contortus* adults were isolated from sheep abomasa six weeks after inoculation using the agar method and manually separated according to sex [24]. Sheep livers were obtained together with abomasa in conditions described previously [18]. In addition, female Wistar Han rats (4–5-month-old) were obtained from the Velaz s.r.o (Prague, Czech Republic).

### 2.3. Egg Hatch Assay

Freshly isolated eggs were treated with nine concentrations of two-fold serial dilution of BLK127, HBK4 or thiabendazole (TBZ, used as the positive control), pre-dissolved in DMSO. The highest tested concentrations were 200, 50, and 15 µM for BLK127, HBK4, and TBZ, respectively. The concentration of DMSO in all samples did not exceed 0.5% *v*/*v*. The control samples contained DMSO only. The eggs were incubated for 48 h in 96-well plates at 27 °C. Each well contained approximately 50 eggs in 200 µL of water solution. The incubation was terminated by adding 5 µL of Lugol’s iodine solution, followed by an enumeration of eggs/larvae under the microscope.

### 2.4. Testing the Viability of Adult H. contortus

The efficacy of BLK127 and HBK4 on *H. contortus* adults (males and females separately) was assessed [23]. The assay was based on the measurement of ATP content as a viability marker after incubation (48 h) of *H. contortus* adults (37 °C, 5% CO_2_) with test compounds. The test concentrations for BLK127 and HBK4 were 1, 10, 20, 30, 40 µM. All test compounds were pre-dissolved in DMSO (<0.1% *v*/*v*). The control samples contained DMSO only. Each concentration was tested on 8 females and 16 males.

### 2.5. Hepatotoxicity Testing

The hepatotoxicity of the BLK127 and HBK4 was evaluated in two animal species (sheep and rat) using two models: PCLS and isolated hepatocytes.

PCLS were prepared using an established method [25]. In brief, after incubation (24 h; 5% CO_2_, 37 °C; humid atmosphere) with the test compounds, the PCLS were washed in PBS, placed into 150 µL of SONOP (sonification solution, 70% ethanol with 2 mM of ethylenediaminetetraacetic acid, EDTA), rapidly frozen on dry ice and stored (−80 °C) until the measurement was taken. PCLS viability was assessed by measuring the ATP level using the manufacturer’s protocol (ATP Bioluminescence assay kit CLS II, Roche, Mannheim, Germany). The concentrations of test compounds were 1, 10, 50, 100 µM for BLK127 and 0.5, 1, 5, 10, 50 µM for HBK4 in the sheep liver models. The concentrations of BLK127 in the rat liver models were the same as for the sheep models. The tested concentrations of HBK4 in the rat model were 1, 10, 50 µM. For HBK4, 50 µM was the maximum concentration due to the limited solubility. All tested compounds were pre-dissolved in DMSO (<0.1% *v*/*v*), and control samples contained DMSO alone.

ATP levels were normalized to the total protein amount (mg). The protein content was measured by bicinchoninic acid assay (BCA) according to the manufacturer’s protocol with the adjustments described previously [23].

Hepatocytes were isolated using a two-step collagenase method [23] and incubated in 96-well plates, precoated with collagen. The density of hepatocytes was 50,000 cells/well. For the experiments, only cell suspensions with >75% of viable hepatocytes were used (assessed in a Trypan Blue exclusion assay–0.4% *w*/*v* Trypan Blue solution). The hepatocytes were incubated with increasing concentrations of BLK127 or HBK4 for 24 h (5% of CO_2_ at 37 °C; humid atmosphere). The concentrations of BLK127 and HBK4 for hepatocytes from rats and sheep were the same as for the PCLS. Immediately after incubation, the viability was measured by adding MTT solution (3 µg/µL, 3-(4,5-dimethylthiazol-2-yl)-2,5-diphenyltetrazolium bromide). After 1 h of incubation, when the viable cells converted yellow, water-soluble MTT to a purple water-insoluble formazan, the cells and formazan crystals were dissolved by replacing of culture medium with isopropanol containing 0.08 M of hydrochloric acid (HCl) and incubated for 30 min (37 °C, 600 rpm, Thermomixer Comfort, Eppendorf). Subsequently, the absorbance was measured at 570 nm, with a background subtraction at 690 nm (Spark Control Tecan, v. 2.2).

### 2.6. Biotransformation of BLK127 in H. contortus

*H. contortus* adults (15 males or 10 females per well) were incubated for 24 h (37 °C, 5% CO_2_; humid atmosphere) in 1.5 mL of RPMI-1640 medium containing 10 µM of BLK127 (pre-dissolved in DMSO) or DMSO only (control) in 24-well plates. The concentration of DMSO in all samples was <0.1%. The RPMI-1640 medium was supplemented according to Kotze and McClure [26] to contain 0.8% of glucose, 0.25 µg/mL of amphotericin B, 10 U/mL of penicillin, 10 µg/mL of streptomycin and 10 mM of HEPES (*N*-[2-hydroxyethyl]piperazine-*N*′-[4-butanesulfonic acid] buffer at pH 6.8). The incubation was ended by the collection of worms and media in separate microtubes. The worms were washed three times in phosphate buffer saline (PBS) before storage (−20 °C).

Prior to the UHPLC-MS analysis, the worms were homogenized using FastPrep-24 5G homogenizer (MP Biomedicals, Irvine, CA, USA), and homogenized samples were cleaned by protein precipitation with ACN. In detail, the worms were homogenized in 1 mL of ultra-pure water using zirconia beads (sizes of 1.0 mm:1.4 mm:2.0 mm in 1:1:0.5 ratio, approx. 0.75 g) by the FastPrep homogenizer (3 cycles: 6 m/s, 20 s). An amount of 700 µL of homogenate was transferred into a 5 mL microtube, after which 1.4 µL IS (250 nM) and ACN (2.1 mL) were added, and the samples were shaken for 1 h. Consequently, the samples were centrifuged (Centrifuge Eppendorf 5810R, 3 min, 3000 rpm (1690× *g*)) and 2.5 mL of supernatant was evaporated (Concentrator Eppendorf plus, Hamburg, Germany, 30 °C) in a silanized vial (Agilent, Santa Clara, CA, USA) and stored until analysis (4 °C). The protein content was measured in the homogenate by a BCA assay according to the manufacturer’s protocol.

The medium samples were extracted by liquid–liquid extraction (LLE). An amount of 3 mL of ethyl acetate was added into 1 mL of media with 1.4 µL IS (250 nM), following which the samples were shaken for 1 h, centrifuged (Centrifuge Eppendorf 5810R, 3 min, 3000 rpm (1690× *g*)) and the upper organic layer was transferred into a silanized vial, evaporated (Concentrator Eppendorf plus, Hamburg, Germany, 30 °C) and stored until analysis (4 °C).

Before the analysis, the samples were reconstituted in 100 µL of 30% ACN (*v*/*v*) and filtrated through a syringe filter (polytetrafluoroethylene 4 mm, 0.22 µm, pk/1000; Labstore, PL70A-102).

### 2.7. Biotransformation of BLK127 in Ovine Liver

Biotransformation of BLK127 was studied in two liver models: PCLS and isolated hepatocytes. PCLS and isolated hepatocytes were incubated for 24 h with 10 µL of BLK127 (or 0.1% DMSO for the control samples) using the conditions described above (Section 2.5). After incubation, the slices were washed in PBS, placed into 200 µL of ultra-pure water and frozen (−20 °C). The isolated hepatocytes were seeded onto a Petri dish (6 cm diameter) precoated with collagen. At the end of the incubation, the hepatocytes were washed in PBS, then mixed with 500 µL of ultra-pure water and collected into 2 mL microtubes. The media from the liver slices and isolated hepatocytes were collected into separate microtubes. Prior to homogenization, the volume of the samples was topped up to 1 mL with ultra-pure water. The homogenization and extraction procedures were the same as for *H. contortus* samples (Section 2.6).

### 2.8. UHPLC-MS and HRMS Conditions

The metabolites were firstly identified using the UHPLC system (the Dionex Ultimate 3000 hyphenated) with the HRMS detector Q Exactive Plus Orbitrap (Thermo Fisher Scientific, Breen, Germany). The UHPLC system consists of the RS LPG quaternary pump, RS column compartment, RS autosampler and Chromeleon software (version 7.2.9, build 11323; Thermo Fisher Scientific, Germering, Germany). The data were analyzed using the Thermo Xcalibur software (version 4.3.73.11).

The sample was injected into the Zorbax Eclipse Plus C18 (2.1 × 150 mm, 1.8 µm, Agilent, Santa Clara, CA, USA) column tempered to 35 °C. The mobile phase consisted of 0.1% formic acid solution (*v*/*v*) in ultra-pure water (A) and ACN (B). The flow rate of the mobile phase was 0.3 mL/min.

The separation method for *H. contortus* samples was performed by gradient elution as follows: 0.0–1.0 min 5% B; 1.0–16.0 min linear gradient to 100% B; 16.0–18.0 min 100% B; 18.0–22.5 min 5% B. The total run time was 22.5 min. The injected sample volume was 10 µL (HRMS method 1).

The method for analysis of ovine liver samples was performed by gradient elution as follows: 0.0–1.0 min 10% B; 1.0–8.0 min linear gradient to 40% B; 8.0–11.0 min linear gradient to 100% B; 11.0–12.0 min 100% B; 12.0–17.0 min 10% B. The total run time was 17.0 min. The injected sample volume was 5 µL (HRMS method 2).

Apart from the metabolite identified as glycine conjugate of hydrolyzed BLK127, which was detected in the negative mode, all other metabolites were detected in the positive mode. The settings of the heated electrospray source were as followed: spray voltage: 4.5 kV; capillary temperature: 270 °C; sheath gas: 45 arbitrary units; auxiliary gas: 12 arbitrary units; spare gas: 2.5 arbitrary units; probe heater temperature: 325 °C; max spray current: 100 µA, S-lens RF level: 50.

Mass spectra were collected at a resolution of 140,000 in the range of 105–800 *m*/*z* in both positive and negative ion modes. All ion fragmentations and parallel reaction monitoring (PRM) were performed at a normalized collision energy of 10–25 with a resolution of 70,000 and a scan range of 60–700 *m*/*z*.

After metabolite identification, the semi-quantification of the metabolite was performed using the UHPLC Nexera liquid chromatograph (Shimadzu, Kyoto, Japan) coupled with a triple quadrupole mass analyser (LC-MS-8030, Shimadzu, Kyoto, Japan).

LabSolution LCMS software 5.93 (Shimadzu, Kyoto, Japan) was used for data analyses. An amount of 2 µL of each sample was injected into the column. The column and mobile phase used were the same as described above. The column temperature was set to 40 °C. The flow rate of the mobile phase was 0.4 mL/min.

For the detection of *m*/*z* 476 and 209 [M+H]^+^, the gradient elution was used as follows: 0.0–6.0 min linear gradient from 30% to 100% of B; 6.0–8.0 min 100% B; 8.0–9.0 min linear gradient to 30% B; 9.0–11.0 min 30% B. The total run time was 11 min (MS method 1).

For detection of *m*/*z* 330 [M+H]^+^ the gradient elution slightly differed and was set as follows: 0.0–12.0 min linear gradient from 10 to 100% of B; 12.0–15.0 min 100% B; 15.0–16.0 min linear gradient to 10% of B; 16.0–18.0 min 10% B. The total run time was 18 min (MS method 2).

Detection was performed in the positive ion mode with the following parameters of electrospray: spray voltage: 4.5 kV; capillary temperature: 250 °C; drying gas: 13 L/min; nebulizing gas: 2.5 L/min; heat block temperature: 400 °C.

Selected ion transitions (selected reaction monitoring, SRM) were used for the detection of the target compounds. Individual SRM with retention times (t_R_) and collision energy (CE) for each compound are listed in Table 1.

### 2.9. Statistical Analyses

The EHT was performed in 5 independent experiments (*n* = 5), each in 3 technical replicates. The viability experiments were performed 3 times in 4 biological replicates in each of the experiments (*n* = 12). Hepatotoxicity tests were performed in 3 independent experiments (*n* = 3), each in 3 technical replicates. One-way ANOVA with Dunnett’s multiple comparison was used for the statistical analysis. All BLK127 biotransformation experiments in the ovine liver and *H. contortus* were performed 3 times (*n* = 3), each in 3 technical replicates. Two-way ANOVA with Tukey’s multiple comparisons test was used for the statistical analysis. GraphPad Prism 9.1.2. was used for all statistical analyses.

## 3. Results

### 3.1. The Effect of BLK127 and HBK4 on the Hatching of H. contortus Eggs

BLK127 and HBK4 had no effect on eggs hatching, even at high concentrations. Used as a positive control, TBZ significantly inhibited egg hatching at concentrations of 1 µM and higher, with IC50 calculated as 1.97 ± 0.03 µM (95% CI: 1.74–2.24 µM).

The results are shown in Figure 1.

### 3.2. The Effect of BLK127 and HBK4 on the Viability of H. contortus Adults

While HBK4 did not show any effect (Figure 2), BLK127 significantly decreased ATP levels at a concentration of 40 µM in females and at 20 µM in males (Figure 3).

Based on these results, we also performed experiments with BLK127 on two resistant strains (IRE and WR) of *H. contortus*. BLK127 significantly lowered the viability of females of the IRE and WR strains at a concentration of 1 µM, and in males of the WR strain at a concentration of 10 µM. For the males of the IRE strain, no significance was observed, probably due to the high variability of the results (Figure 3).

### 3.3. Hepatotoxicity of BLK127 and HBK4

In all experiments, BLK127 showed no negative impact on any of the liver models (Figure 4). In contrast, HBK4 significantly decreased the viability of sheep and rat PCLS at concentrations of 10 and 1 µM, respectively, and at 5 µM in isolated hepatocytes from both animals (Figure 5).

### 3.4. Biotransformation of BLK127 in H. contortus

Three metabolic pathways were identified in *H. contortus* adults: hydroxylation, hydrolysis and glycosidation. Metabolites were identified based on accurate mass, retention times (t_R_) and fragmentation spectra. The metabolites that were identified in the real samples were absent from the blank samples.

The parent compound BLK127 *m*/*z* 314.1751 [M+H]^+^, eluted in t_R_ 15.64 min, was characterized by product ions at *m*/*z* 191.1067, 121.0289 and 91.0548 (Appendix A). Two of the most abundant metabolites at *m*/*z* 476.2279 [M+H]^+^, detected in t_R_ 10.99 and 12.37 min, contained the characteristic product ions of BLK127 in their MS/MS spectra; therefore, the metabolite originates from BLK127. Due to the mass shift of *m*/*z* 162, typical for glucose conjugation [27], we tentatively identified this metabolite as BLK127-N-glycoside (BLK127-N-glc, Appendix A). We were not able to identify the particular hexose; however, the discovery of the two metabolites suggests conjugation with two different ones. Moreover, conjugation with glucose as well as other hexoses in *H. contortus* has also been reported previously [28].

The second most abundant metabolite at *m*/*z* 209.1169 [M+H]^+^, formed by adult worms, was identified as hydrolyzed BLK127 (h-BLK127). The main product ions of h-BLK127 at *m*/*z* 191.1064 and 121.0284 correspond to the product ions of BLK127. The less abundant h-BLK127 fragment at *m*/*z* 139.0388 [M+H]^+^ is created by cleavage of the five-carbon chain. The fragment at *m*/*z* 95.0496 [M+H]^+^ corresponds with *m*/*z* of hydroxybenzene (Appendix A). The h-BLK127 was also detected in blank chemical samples; however, its intensity was ~ 10-times less than in the samples.

Moreover, three isomers at *m*/*z* 330.1696 [M+H]^+^, identified as hydroxylated BLK127 (BLK127-OH), were detected in t_R_ 11.24, 11.78 and 12.05 min. The product ions at *m*/*z* 207.1169 and 121.0285 suggest that the hydroxyl group is bonded to an aromatic circle of the 4-(pentyloxy)benzamide part of the molecule (Appendix A). Additionally, three O-glycoside metabolites of BLK127 (BLK127-O-glc) were detected at t_R_ 9.20, 9.60 and 9.85 min. BLK127-O-glc was identified based on the accurate mass of the precursor ion *m*/*z* 492.2224 [M+H]^+^. Due to the low abundance of these metabolites, we were not able to identify the main fragments (Appendix A).

All the metabolites were detected in both the culture medium and worms; however, the intensity of the metabolites detected differed between these compartments. While BLK127-N-glc was detected at a similar intensity in worms and media, BLK127-OH was predominant in the medium, and the findings of h-BLK127 were inconsistent between samples.

Additionally, we compared the extent of biotransformation between sensitive and resistant strains of *H. contortus*. The metabolites were semi-quantified using the peak area ratio normalized to the total protein amount (mg). Although females of the WR strain produced significantly more BLK127-N-glc than females of the ISE strain, an opposite trend was seen in the males, such that, in general, the biotransformation of BLK127 between sensitive and resistant strains did not differ markedly (Figure 6).

The proposed metabolic pathways are presented in Figure 7.

### 3.5. Biotransformation of BLK127 in Ovine Liver

PCLS and isolated hepatocytes metabolized BLK127 almost completely, compared with *H. contortus*, in which most of the parent drug remained unmetabolized. Compared with the parent compound, BLK127 was not detectable in the ovine liver samples after incubation for 24 h.

Ovine liver metabolized BLK127 by hydrolysis, forming the 4-(pentyloxy)benzoic, subsequently conjugated with glycine (h-BLK127-gly). The glycine conjugation of h-BLK127 was identified based on the accurate mass at *m*/*z* 264.1241 [M-H]^−^ and a characteristic loss of CO_2_ in the negative mode (Appendix A). In both liver models employed, this metabolite was present in the homogenate and the medium. The proposed metabolic pathways are described in Figure 8.

The detailed information (elemental composition, designation, *m*/*z* of precursor and product ions, collision energy (CE), t_R_ about BLK127, the IS, and the metabolites detected in *H. contortus*, PCLS and isolated hepatocytes) are given in Table 2.

The HRMS spectrum of IS is shown in Appendix A.

## 4. Discussion

Nguyen et al. [10] screened compounds in the “Kurz-Box” and revealed two compounds—BLK127 and HBK4—both of which showed a significant adverse effect on the larval stages of *H. contortus*. However, nothing was known about their effects on the eggs and adults of *H. contortus*, nor about their hepatotoxicity and biotransformation. The present study tackled these areas.

Although the egg stage is usually not the main target for anthelmintic treatment, EHT continues to be used for assessing the ovicidal activity of new nematocides [29]. Despite this, our results showed that none of the tested compounds had any adverse effect on egg hatching.

Subsequently, we exposed adults of *H. contortus* to BLK127 and HBK4 for 48 h and measured their viability via ATP bioluminescent assay [23,30]. The results showed that BLK127 significantly decreased the viability of females and males of the ISE strain at 40 µM and 20 µM, respectively. A slight decrease in viability was also observed at 30 μM in males after HBK4 treatment, although due to variability among samples, this decrease was not statistically significant. In addition, the effect of BLK127 on the IRE and WR strains of *H. contortus* was compared, with BLK127 decreasing viability in the resistant strains from 1 μM. In comparison to this result, in previous work [10], BLK127 and HBK4 inhibited L4 motility at concentrations from 12.5 μM and 25 μM, with IC_50_ values at 7.45 ± 1.76 µM and 12.17 ± 2.28 µM, respectively.

We also focused on evaluating the toxicity of BLK127 and HBK4 in the rat and sheep liver models. While BLK127 showed no toxicity in any of the liver models, even at the highest tested concentration, HBK4 was shown to be toxic to all the liver models, even at a concentration of 10 μM. These findings suggest that HBK4 is not a suitable candidate for future testing. Therefore, only BLK127 was investigated further regarding biotransformation in adults of *H. contortus* and sheep liver.

For a long time, it was thought that oxidative metabolism is not present in helminths; however, recent studies provide evidence to the contrary [22,31]. The formation of BLK127-OH in this work also supports the presence of oxidative metabolism in helminths. Hydroxylation has also been described as the main metabolic pathway in sertraline [23]. In addition, we identified three other metabolites: hydrolyzed BLK127, BLK127-*N*-glycoside and BLK127-*O*-glycoside. Each of these metabolites was found in all three strains of *H. contortus*, although most of the parent compound remained unmetabolized. Glucosidation represents the most important pathway in plants and invertebrates [27], and the formation of the *N*- and *O*-glycosides of various anthelmintics has also been reported previously in *H. contortus* [21,28].

Here, the only metabolite of BLK127 detected in the sheep liver models was the glycine conjugate of the h-BLK127 intermediate. Conjugation with glycine represents the detoxification reaction of carboxylic acids and it is often observed in sheep, cats and gerbils, and it was the first biotransformation reaction discovered, in 1842 [32,33].

To further support our results, we compared the properties of both compounds according to Lipinski’s rule of five, used as a guide for rational drug design (Table A1). The results show that BLK127 meets all five rules, while HBK4 did not meet the rule for molecular weight. Considering that BLK127 satisfies all the parameters of this rule, this suggests that it might be administrable orally [34].

## 5. Conclusions

In conclusion, compound BLK127 from the “Kurz-box” showed several promising characteristics. It was effective against *H. contortus—*including resistant strains—it was not hepatotoxic and was shown to form several metabolites in a biotransformation study, albeit at low quantities. This suggests that *H. contortus* cannot protect itself well against the compound effect through metabolism. These findings suggest that BLK127 is a promising new anthelmintic candidate.

## Figures and Tables

**Figure 1 pharmaceutics-14-00754-f001:**
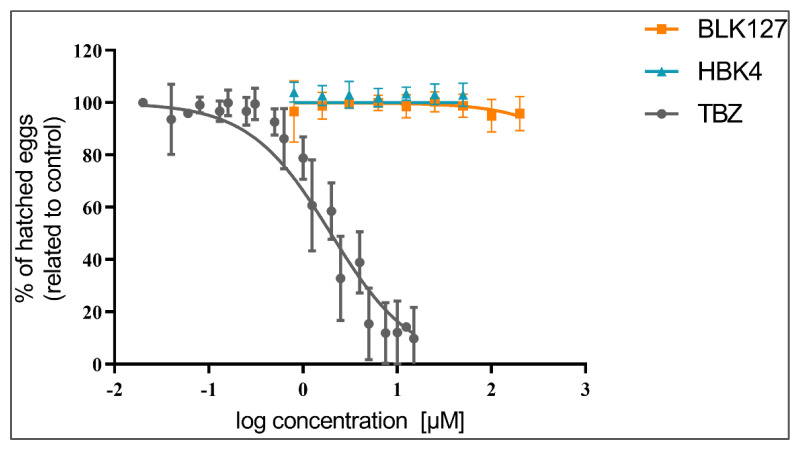
Effect of compounds BLK127, HBK4 and TBZ on *H. contortus* egg hatching (ISE strain) in comparison to a negative (no-compound) control (0.5% *v*/*v* DMSO). The data are presented as a means ± SD (*n* = 5).

**Figure 2 pharmaceutics-14-00754-f002:**
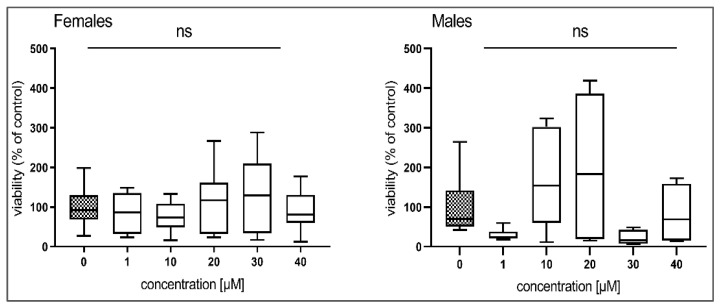
The effect of compound HBK4 on the viability of *H. contortus* adults of the ISE strain after incubation (48 h). The data are expressed in percentage of viability of the controls (=100%) and presented as a box plot (*n* = 12). For statistical analysis, one-way ANOVA with Dunnett’s multiple comparison test was used. ns, *p* > 0.05.

**Figure 3 pharmaceutics-14-00754-f003:**
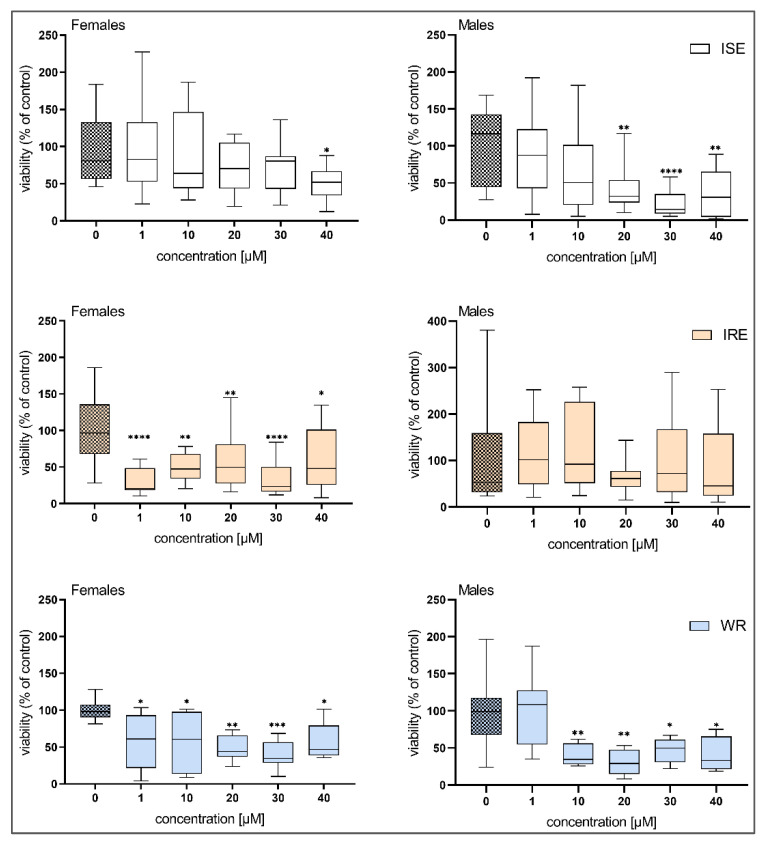
The effect of compound BLK127 on the viability of *H. contortus* adults from the ISE, IRE and WR strains after incubation (48 h). The data are expressed in percentage of viability of the controls (100%) and presented as a box plot (*n* = 12). For statistical analysis, one-way ANOVA with Dunnett’s multiple comparison test was used. *, *p* < 0.05; **, *p* < 0.01; ***, *p* < 0.001; ****, *p* < 0.0001.

**Figure 4 pharmaceutics-14-00754-f004:**
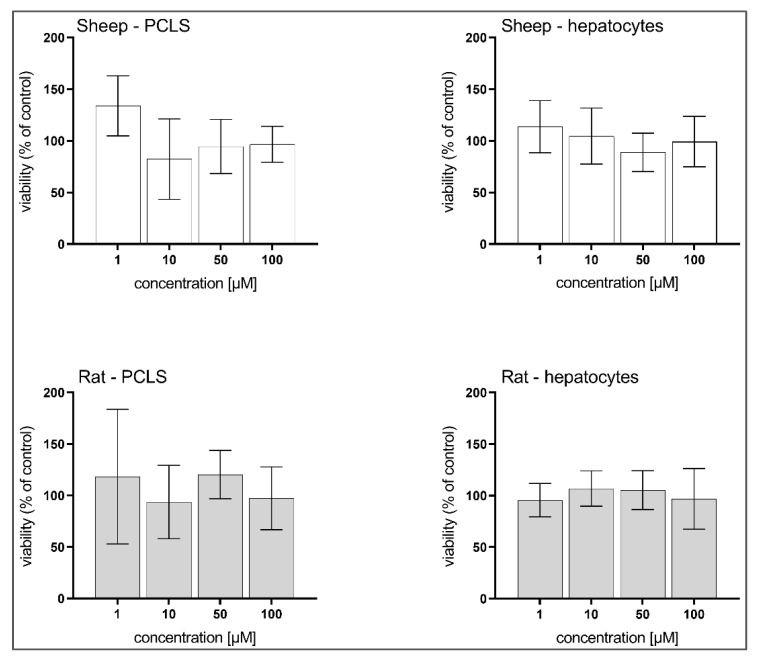
Effect of compound BLK127 on the viability of precision-cut liver slices (PCLS) and isolated hepatocytes of ovine and rat. The data are expressed in percentage of viability of the controls (100%) and as means ± SD (*n* = 3). For statistical analysis, one-way ANOVA with Dunnett’s multiple comparison test was used.

**Figure 5 pharmaceutics-14-00754-f005:**
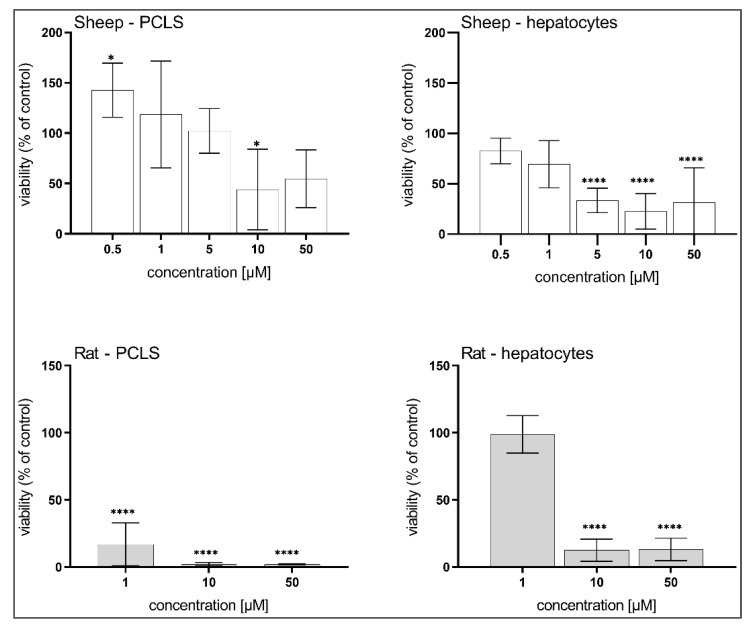
Effect of compound HBK4 on the viability of precision-cut liver slices (PCLS) and isolated hepatocytes of ovine and rat. The data are expressed in percentage of viability of the controls (=100%) and as means ± SD (*n* = 3). For statistical analysis, one-way ANOVA with Dunnett’s multiple comparison test was used. *, *p* < 0.5; ****, *p* < 0.0001.

**Figure 6 pharmaceutics-14-00754-f006:**
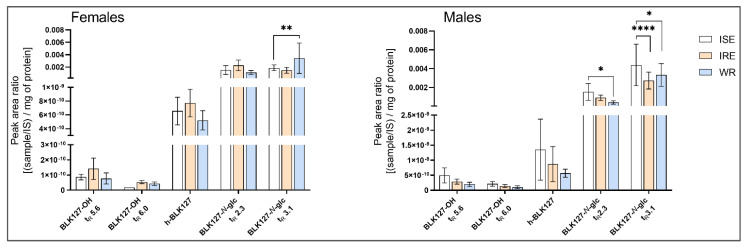
The amounts of compound BLK127 metabolites formed in adults of *H. contortus* from ISE, IRE and WR strains. Semi-quantification was performed using the peak area ratio normalized to the total protein amount (mg). Data are presented as means ± SD. For statistical analysis, two-way ANOVA with Tukey’s multiple comparisons test was used. *, *p* < 0.05; **, *p* < 0.01; **** *p* < 0.0001.

**Figure 7 pharmaceutics-14-00754-f007:**
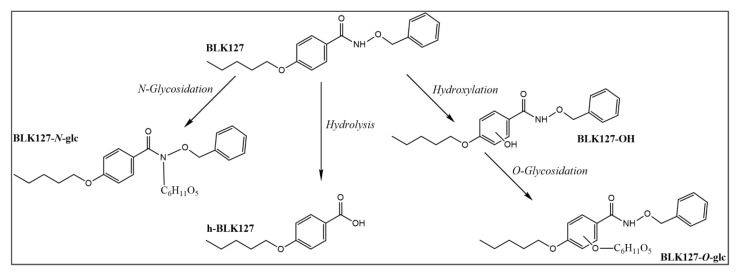
The proposed metabolic pathways of compound BLK127 in *H. contortus* adults.

**Figure 8 pharmaceutics-14-00754-f008:**
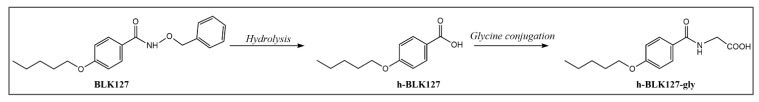
The proposed metabolic pathways of compound BLK127 in the ovine liver.

**Table 1 pharmaceutics-14-00754-t001:** List of the *m*/*z* of precursor ions, retention times (t_R_), selected reaction monitoring (SRM), and collision energy (CE) for BLK127, internal standard and its main metabolites in *H. contortus* adults measured by LC-MS.

*m*/*z* Precursor Ions [M+H] ^+^	t_R_ LC-MS [min]	SRM [M+H] ^+^	CE
314	5.0 ^1^; 8.9 ^2^	190.95	−11.0
		90.95	−25.0
		121	−26.0
322	4.8 ^1^; 8.5 ^2^	198	−13
		91	−12
		191	−22
209	2.8 ^1^	121	−20
		139	−20
330	5.6 ^2^	207	−20
	6.0 ^2^	121	−20
	6.2 ^2^	91	−20
476	2.3 ^1^	191	−20
	3.1 ^1^	121	−20
		91	−20

^1^ MS method 1. ^2^ MS method 2.

**Table 2 pharmaceutics-14-00754-t002:** List of the main metabolites of BLK127 detected in *H. contortus* adults and in the ovine liver, their elemental composition, designation, retention times (t_R_), *m*/*z* of the precursor and product ions, and collision energy (CE) detected by LC-HRMS. The table includes the parent drug BLK127 and the internal standard (IS) information.

Compound	Elemental Composition	Designation	tR LC-HRMS [min]	*m*/*z* Precursor Ions [M+H]^+^ [M−H]^−^	*m*/*z* Product Ions [M+H]^+^ [M−H]^−^	CE	Occurrence
BLK127	C_19_H_23_NO_3_	-	15.64 ^1^; 14.22 ^2^	314.1751	191.1063	20	-
					121.0285		
					91.0547		
Internal standard	C_20_H_19_NO_3_	IS	15.24 ^1^; 14.11 ^2^	322.1431	199.0752	20	*-*
					171.0802		
					127.0549		
					91.0548		
BLK127 hydrolysis	C_12_H_16_O_3_	h-BLK127	13.971	209.1169	191.1064	20	*H. contortus*
					139.0388		
					121.0284		Ovine liver
					95.0496		
Hydroxylated BLK127	C_19_H_23_NO_4_	BLK127-OH	11.24 ^1^	330.1702	207.1015	25	*H. contortus*
			11.78 ^1^		121.0285		
			12.05 ^1^		91.0548		
BLK127 *N*-glycosidation	C_25_H_33_NO_8_	BLK127-*N*-glc	10.99 ^1^	476.2279	268.1176	25	*H. contortus*
			12.37 ^1^		232.0977		
					209.117		
					191.1065		
					121.0285		
					91.0548		
BLK127 *O*-glycosidation	C_25_H_33_NO_9_	BLK127-*O*-glc	9.20 ^1^	492.2224	-	15; 25	*H. contortus*
			9.60 ^1^				
			9.83 ^1^				
Hydrolysed BLK127-glycine	C_14_H_19_NO_4_	h-BLK127-gly	10.572	264.1246	220.1335	20	Ovine liver
					93.03333		

^1^ HRMS method 1. ^2^ HRMS method 2.

## Data Availability

The data presented in this study are included within this article and Appendix A.

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
