# Peer review of "Assessing the Anthelmintic Candidates BLK127 and HBK4 for Their Efficacy on *Haemonchus contortus* Adults and Eggs, and Their Hepatotoxicity and Biotransformation"

_pharmaceutics, 2022, doi:10.3390/pharmaceutics14040754_

Round 1
Reviewer 1 Report
The authors evaluate the efficacy of candidate anthelmintics BLK127 and HBK4 to reverse Haemonchus adults and eggs, as well as their hepatotoxicity and biotransformation. This research is meaningful and original. The method is complete and novel, so I recommend accepting it in its current form. The only regret is that I didn't see the author mentioning figure S1-7, please upload it in time.
Author Response
Dear reviewer, thank you very much for reading and evaluation of our MS. The supplementary files were uploaded.
Reviewer 2 Report
This manuscript shows interesting results about the anthelmintic efficacy of two compounds against eggs and adults Haemonchus contorts, one of the most pathogenic parasitic nematodes for small ruminants all over the world. Additionally, the study shows complementary information about the hepatotoxicity in sheep and rats and biotransformation of one of these compounds in H. contortus adults and in ovine liver. The manuscript is well structured and in general it has a very clear scientific description of its objectives, methodological strategy and results. I had a problem to access to the UHPLC chromatograms (Supplementary materials) (Figures S1 to S7). I was not able to access to these figures. When I attempted to download these files a window indicated me that it was an ERROR 404 -FILE NOT FOUND. Regarding the source of obtaining the 236 chemical compounds for screaming “Kurz Box”; I think authors could briefly describe more information about “Kunz Box”, mainly for people like me that are not familiarised with this concept. After I checked the whole manuscript, I found only a few details that will have to be fixed and that I have marked in yellow colour in the manuscript. Regarding their results, it seems that they found a very promising compound candidate to be explored as a potential anthelmintic compound. This issue is important because the anthelmintic resistance in the parasites against most of commercially available anthelmintic compounds in small and large ruminants is dramatically increasing in the whole world and this problem continuously threaten the animal health and production in any country. Searching for new strategies of control and new anthelmintic molecules is crucial for facing the problem of gastrointestinal parasitic nematodes. In my opinion, the manuscript presented contains an important package of solid information that open new expectations for a new potential anthelmintic compound that will have to be explored for the control of H. contortus and perhaps against other parasites of this group of pathogens. I think this manuscript should be accepted for publications after MINOR REVISION.

Author Response
Dear reviewer, thank you very much for reading, evaluation, and correction of our MS. According to your suggestions we made the following amendments:
- The supplementary files were uploaded.
- More information about “Kurz-box” was added in the Introduction (line 49) and the designation “Kurz box” was removed from the Abstract
- Line 43: The space was added between the words.
- Line 72: „Was“ was changed to „were”.
- Line 73 and 74: The information about strains of H. contortus was added.
- Line 77: The sentence was rewritten.
- Line 114: The information about number of treatment and the volume in the wells were added.
- Line 158: The information about number of replicates is included in subsection “2.9. Statistical analyses”.
- Line 268: The text was removed from the result section.
Reviewer 3 Report
The article is a continuation of the research by Nguyen et all. (2019). The manuscript provides interesting and practical data on the effect of the compounds BLK127 and HBK4, especially BLK127 on the nematode Haemonchus contortus (eggs and adults).
Undoubtedly, the conducted research (results) may be helpful in the treatment of the haemonchosis, although further studies are needed.
I am of an opinion that the article fits into scope of Pharmaceutics and could be published after some minor corrections.
- Line 94: “2.2. Parasites and animals” – maybe “Parasites and hosts” ?. Nematodes are also animals.
- Line 102: Whether the flotation and centrifugation used did not damage the eggs ?
- Line 106: Why female rats were used in this study ? I am asking because males are more often used in many studies.
- Line 109: “Freshly isolated eggs were treated…” – of course, it is about eggs isolated by the flotation method (line 102) ?.
- Lines 116-122: How many adult nematodes were used in each test.
- Lines 262-263: “Freshly isolated eggs were incubated with BLK127, HBK4, and TBZ for 48 h. After incubation, the numbers of hatched larvae and unhatched eggs were counted” – this belongs to the Material and methods.
- Lines 272-273: The effects of BLK127 and HBK4 were assessed on the ISE strain of H. contortus. The level of ATP was used as a viability marker after incubation (48 h)” – this belongs to the Material and methods.
- Figure 2: What does the symbol "ns" mean
- Lines 286-287: “The hepatotoxicity of BLK127 and HBK4 was tested in sheep and rat PCLS as well as isolated hepatocytes” - this belongs to the Material and methods.
- References list:
All scientific names (genus, species) must be italicized.
Journal names should be abbreviated.
Author Response
Dear reviewer, thank you very much for reading, evaluation, and correction of our MS. According to your suggestions we made the following amendments:
Line 94: “2.2. Parasites and animals” – maybe “Parasites and hosts” ?. Nematodes are also animals.
- The title of the paragraph was rewritten.
Line 102: Whether the flotation and centrifugation used did not damage the eggs?
- This procedure has been used commonly for eggs isolation and it don’t cause the eggs damage. Confirmation of eggs vitality is their hatching in the control after 48 hour-incubation.
Line 106: Why female rats were used in this study? I am asking because males are more often used in many studies.
- Because we had the females available.
Line 109: “Freshly isolated eggs were treated…” – of course, it is about eggs isolated by the flotation method (line 102)?
- The eggs were isolated by sucrose flotation procedure. There is no need to repeat the information as the information about eggs isolation is clearly mentioned in the paragraph “2.2. Parasites and host”.
Lines 116-122: How many adult nematodes were used in each test.
- The information was added.
Lines 262-263: “Freshly isolated eggs were incubated with BLK127, HBK4, and TBZ for 48 h. After incubation, the numbers of hatched larvae and unhatched eggs were counted” – this belongs to the Material and methods.
- The sentences were removed from the paragraph.
Lines 272-273: The effects of BLK127 and HBK4 were assessed on the ISE strain of H. contortus. The level of ATP was used as a viability marker after incubation (48 h)” – this belongs to the Material and methods.
- The sentence was removed from the paragraph.
Figure 2: What does the symbol "ns" mean
- Description of abbreviation was added to the figure description.
Lines 286-287: “The hepatotoxicity of BLK127 and HBK4 was tested in sheep and rat PCLS as well as isolated hepatocytes” - this belongs to the Material and methods.
- The information was removed from the method.
References list:
All scientific names (genus, species) must be italicized.
- Scientific names were italicized.
Journal names should be abbreviated.
- The Journal names were abbreviated
Reviewer 4 Report
The manuscript has presented the effect of compounds (BLK127 and HBK4) against H. contortus eggs as well as adults, and their hepatotoxicity and biotransformation. They have found that BLK127 would be a great candidate for a new anthelmintic based on its high sensitivity and resistance. The manuscript is well organized and provided various precious data to prove their finding. Therefore, I strongly recommend accepting the manuscript for publication in a present form.
Author Response
Dear reviewer, thank you very much for reading and evaluation of our MS.